# Principles of Drug Dosing in Sustained Low Efficiency Dialysis (SLED) and Review of Antimicrobial Dosing Literature

**DOI:** 10.3390/pharmacy8010033

**Published:** 2020-03-09

**Authors:** Paula Brown, Marisa Battistella

**Affiliations:** 1Pharmacy Department, University Health Network, Toronto, ON M4G 2C4, Canada; Paula.Brown2@uhn.ca; 2Leslie Dan Faculty of Pharmacy, University of Toronto, Toronto, ON M5S 3M2, Canada

**Keywords:** acute kidney injury, AKI, sustained low efficiency dialysis, sled, pharmacokinetics, antimicrobials

## Abstract

The use of sustained low-efficiency dialysis (SLED) as a renal replacement modality has increased in critically ill patients with both acute kidney injury (AKI) and hemodynamic instability. Unfortunately, there is a paucity of data regarding the appropriate dosing of medications for patients undergoing SLED. Dose adjustment in SLED often requires interpretation of pharmacodynamics and pharmacokinetic factors and extrapolation based on dosing recommendations from other modes of renal replacement therapy (RRT). This review summarizes published trials of antimicrobial dose adjustment in SLED and discusses pharmacokinetic considerations specific to medication dosing in SLED. Preliminary recommendation is provided on selection of appropriate dosing for medications where published literature is unavailable.

## 1. Principles of Drug Dosing in Sustained Low Efficiency Dialysis (SLED)

Intermittent hemodialysis (IHD) or continuous renal replacement therapy (CRRT) has been provided to critically ill patients with acute kidney failure. IHD is often complicated by hypotension and inadequate fluid removal [1]. Although CRRT addresses some of the shortcomings of IHD, it is associated with significantly greater complexity, the need for continuous anticoagulation, and higher costs [2]. Over the past ten years, slow extended daily dialysis (SLEDD) has become an alternate modality to CRRT [3,4,5]. It represents a “hybrid” that uses IHD and CRRT. Hybrid therapies are also known as prolonged intermittent renal replacement therapy (PIRRT), sustained low-efficiency dialysis (SLED), and extended daily dialysis (EDD). The increased types of dialysis methods has generated confusion about what is being accomplished during each of these procedures (Table 1). For the purpose of this paper, we will refer to this hybrid as SLED.

Sustained low efficiency dialysis (SLED) has several advantages: First, as with other continuous replacement therapies, SLED has stable hemodynamics secondary to decreased ultrafiltration rate, and low solute removal. Furthermore, the dialysis dose can be increased because of extended treatment duration; and it allows for patients to undergo other investigations or treatments between dialysis sessions [6,7]. SLED is cheaper compared to other CRRT [7]. For instance, conventional dialysis machines are used and so no additional equipment is needed [7]. The dialyzers are also inexpensive and a standard dialysate concentrate is used, rather than specialized dialysate or ultrafiltrate replacement solutions [7]. Moreover, anticoagulation is not generally used for SLED [7]. Some differences between SLED and other traditional continuous therapies include: clearances for small molecules are generally higher per hour than they are in CRRT; keeping in mind that SLED is generally used for only 6–12 h per day (compared to 24 h) and therefore overall clearance remains relatively the same [8]. Furthermore, with SLED, there may be less removal of middle-sized molecules (i.e., 1000–10,000 Daltons) compared to CRRT as the dialyzer membrane used in SLED are generally less permeable [7]. SLED has some disadvantages such as unfamiliarity with the modality, and hypophosphatemia [7].

Therefore, with the increasing use of SLED as a modality for critically ill patients in the ICU, it is essential that clinicians understand the pharmacokinetic and pharmacodynamic properties of medications, in order to help them make informed decisions on optimum therapy. In this article, we discuss the general pharmacokinetic and pharmacodynamic concepts in patients with kidney failure, and the various dosage adjustments that need to be made for specific antibiotics during SLED.

## 2. Pharmacokinetic and Pharmacodynamic Principles during SLED

The pharmacokinetic parameters absorption, distribution, and clearance of medications are altered in critically ill patient population with acute kidney injury receiving SLED; thus these parameters need to be considered when making drug dosing adjustments.

With the uncertainty of oral absorption in critically ill patients, medications are usually administered intravenously. However, if oral medications are given, absorption may be decreased secondary to uremic toxins present [9].

A drug’s volume of distribution (*V*_D_) describes the extent of distribution throughout the body. The *V*_D_ of many drugs is increased in patients with AKI and can lead to a reduction in serum drug concentrations [10]. This increase in *V*_D_ may be the result of pathophysiologic alterations in body composition, fluid overload secondary to excessive fluid administration (and no clearance function), decreased protein binding, or increased tissue binding. Many critically ill patients receive large volumes of intravenous fluids for resuscitation from shock, and can subsequently develop edema, pleural effusions, or ascites. These therapeutic interventions, in addition to reduced water excretion because of AKI (or CKD), often lead to an increase in a drug’s *V*_D_ and a decrease in its serum concentrations. This is especially problematic with hydrophilic drugs, such as aminoglycosides and cephalosporins for which the *V*_D_ may be increased by up to 150% [11].

As only unbound or “free” drug is able to cross cellular membranes and distribute outside the vascular space, protein binding will then limit drug distribution. Recall that most drugs are bound to albumin or alpha-1 acid glycoprotein. Many drugs have been reported to exhibit altered protein binding in critically ill patients secondary to hypoalbuminemia, qualitative changes in the conformation of the protein binding site, and/or competition for binding sites by other drugs, metabolites, and endogenous substances [10,12]. Therefore, as a result of a decrease in protein binding, there is an increase in the apparent *V_D_* [12]. For example, the protein binding of many acidic drugs such as penicillins, cephalosporins, aminoglycosides, furosemide, and phenytoin is reduced in AKI or CKD patients [10,12]. Since the protein binding of these drugs is reduced, this leads to a greater distribution into the interstitial space and thus a potential increased clearance by the liver, kidneys, and/or RRT.

The patient’s residual kidney function and the mode of renal replacement therapy will determine the clearance of a renally eliminated drug. Drug clearance per hour is usually highest with SLED followed by CRRT and then lowest with IHD [13]. However, because the duration of dialysis in SLED is shorter (usually 6 to 12 h per day vs. 24 h per day with CRRT), the overall drug clearance per day is usually less in SLED compared to CRRT but greater than IHD. Finally, drug clearance from dialysis will also be determined by the dialysis prescription which includes dialysate and blood flow rates as well as the type of dialyzer used.

Drug clearance is also influenced by the mechanisms of solute removal, such as convection and diffusion which occurs in SLED [12]. Rates of convection depend on the rate of ultrafiltration, which is dependent on the transmembrane pressure gradient created by the blood and ultrafiltrate pumps in the dialysis system. Diffusion of solutes (or medications) across the hemofilter membrane is dependent on the transmembrane concentration gradient of that solute (or medication). Therefore, diffusion is the most effective method for removal of small molecules (<1000 daltons) and therefore, medications smaller than 1000 daltons will be eliminated by diffusion [13]. Nonrenal drug clearance may also be affected in patients with acute kidney injury. There is some data from human studies suggesting that hepatic drug metabolism and transporter function are affected by AKI [14]. Furthermore, AKI may also impair the clearance of parent drug metabolites. Unfortunately the mechanism of how AKI affects drug metabolism and nonrenal clearance is limited and how to dose adjust to account for this nonrenal clearance is still difficult to predict in patients with AKI. Nevertheless, clinicians should note that even drugs and drug metabolites that have been hepatically removed can accumulate in AKI, and renal replacement therapy may affect nonrenal clearance as well as drug metabolite clearance. Herefore, drug dosing should be reassessed with any changes in kidney function as well as RRT.

The timing of drug administration with respect to the start and duration of SLED will affect overall drug exposure [15]. Because SLED is delivered for over 6–12 h per day and antibiotics are usually administered intermittently, the timing and dosing of medications need to be considered in relation to the timing of SLED. If, however, a medication is administered as a continuous infusion, it would need to have the infusion rate adjusted if the patient is on SLED vs. off SLED. Therefore, when interpreting published pharmacokinetic studies for antibiotics in patients receiving SLED it is important to consider the exact prescription and timing of antibiotic administration. Finally, overall drug elimination can change on a daily basis in the critically ill population as a patient’s residual kidney function may improve or decline. Thus the patient’s residual kidney function should always be considered when making drug dosing adjustments.

In patients receiving RRT, the pharmacodynamic profile of an antibiotic, whether its antimicrobial activity is concentration- or time-dependent, can affect the dosing regimen. For instance, antibiotics such as aminoglycosides and quinolones are considered concentration–dependent antibiotics and are often given less often but at a higher dose. For these concentration dependent antibiotics, a higher concentration relative to the minimum inhibitory concentration (MIC) of the organism results in greater antimicrobial efficacy. On the other hand, beta lactams (i.e., cephalosporins, carbapenems, penicillins) are considered time-dependent antibiotics meaning that their most effective antimicrobial activity is the percentage of time the drug concentration is above the MIC of the organism [13,16]. These antibiotics are often given more frequently and sometimes even continuously. Therefore, when considering the timing of antibiotic administration relative to the initiation and duration of SLED, clinicians must take into account whether the antibiotic is time- or concentration dependent to balance efficacy and toxicity concerns [13,16]. 

Therefore, when evaluating the literature on antibiotic dosing during SLED in order to determine the best dosing regimen, it is important to consider the dialysis modality including dialysate and blood flow rates, the dialyzer type, dialysis duration and frequency, and the patient’s clinical picture [17]. 

## 3. Published Studies of Antimicrobials in Patients Receiving Extended Modes of Dialysis

There have been several small PK studies published in the past one to two decades attempting to determine the PK impact of these extended modes of dialysis and to identify optimal antimicrobial dosing to maximize efficacy while minimizing toxicity [8,9,10,11,12,13,14,15,16,17,18,19,20,21,22,23,24,25,26,27,28,29,30,31,32,33,34,35,36,37,38,39,40,41,42,43,44,45,46,47]. Historically this was done by examining the blood concentrations at multiple time points before, during, and after drug administration. Drug concentrations were assessed using methods such as high-performance liquid chromatography or mass spectrometry. More recently, there have been a few publications of *in silico* pharmacokinetic and pharmacodynamic analyses utilizing Monte Carlo simulations (MCS) and virtual subjects to simulate real world patient populations [21,22,24,26,27,33,35,38,47]. When performing these simulations, researchers can preset a desirable probability of target attainment (PTA) of having a drug concentration greater than a multiple of the MIC for a preset amount of time. For example, the goal may be to have a 90% probability of spending at least 60% of time in the dosing interval with a drug concentration greater than 4 × MIC. This means that theoretically 90% of patients will meet the preset pharmacodynamic targets. Four times the MIC is selected as the target for antibiotics to have maximal bactericidal activity and suppression of bacterial resistance as this is pivotal in treating critically ill patients. Various dosing regimens and dialysis settings can be modeled in MCS in an attempt to determine the regimen that best meets the preset parameters without requiring large numbers of actual subjects, extensive blood sampling, and drug concentration assays. See Table 2, Table 3, Table 4, Table 5, Table 6, Table 7 and Table 8 for a summary of all published trials and case reports identified in a literature review. For the purposes of simplicity, all extended dialysis modes will be referred to as SLED in the following summaries.

### 3.1. Penicillins

#### 3.1.1. Penicillin G

The pharmacokinetics of benzylpenicillin (Penicillin G) in SLED was reported in two critically ill patients. Both patients had penicillin-sensitive *Staphylococcus aureus* bacteremia complicated by infective endocarditis [18]. Dialysis was performed for 9.5 h and 8.5 h in these patients, respectively. Patients were given Penicillin G 1800 mg (3 million units) intravenously over 1 h (patient 1) and 5 min (patient 2) every 6 h on dialysis days. A dose was given within an hour of starting SLED and within an hour after stopping SLED. Multiple blood samples were taken over a 2-day period which included one SLED session. This dosing strategy was able to achieve plasma concentrations of 4–5 times the MIC for 100% of time. No adverse events related to Penicillin G were reported. Patient 1 died on day 25 in the ICU because of overwhelming sepsis and patient 2 was successfully discharged from ICU [18]. 

#### 3.1.2. Ampicillin/Sulbactam

A case was published of a patient that received a single dose of ampicillin/sulbactam for an *Enterococcus faecalis* (MIC < 2) urinary tract infection while on SLED [19]. A dose of 3 g (ampicillin 2 g/sulbactam 1 g) was administered over 30 min and multiple plasma samples were drawn prior to the dose and up to 12 h after the initiation of the infusion. Dialysis was initiated 4 h after the dose was administered and continued for 7.5 h. The patient’s clinical symptoms improved and inflammatory markers normalized. However, it was found the MIC_90_ was only >2 mg/L for 8 h (~30% of the dosing interval) after the start of the infusion. Since time-dependent antimicrobials require 40–50% of the dosing interval to be above the MIC, this dosing is considered insufficient. It is difficult to extrapolate to multiple doses and dialysis sessions, as this case report was only a single dose of ampicillin/sulbacatam and session of SLED. 

A second study included twelve critically ill patients with anuric AKI receiving 8-h SLED given a single dose of ampicillin/sulbactam (2 g/1 g) infused over 30 min [20]. Three of these patients received 4 days of twice-daily ampicillin/sulbactam (2 g/1 g) to study multiple dose pharmacokinetics. Dialysis was started approximately 3 h after the dose of ampicillin/sulbactam. The half-lives were found to be 2.8 ± 0.8 h and 3.5 ± 1.5 h for ampicillin and sulbactam, respectively. This is compared to healthy subjects of 1.41 ± 0.65 h and 1.73 ± 0.72 h. Removal of amipicillin was 87% and sulbactam 93% after a single SLED session. No significant accumulation in patients receiving multiple doses was observed. The authors recommend a dose of ampicillin/sulbactam of at least 2 g/1 g IV every 12 h with one dose given post-dialysis in patients receiving 8-h of dialysis [20].

#### 3.1.3. Piperacillin/Tazobactam

A MCS was performed in critically ill patients receiving SLED targeting free concentrations of piperacillin-tazobactam greater than four times the MIC of *Pseudomonas aeruginosa* for >50% the first 48 h of antibiotic therapy [21]. Four different models of dialysis were simulated [8 h per day (ultrafiltration rate/dialysate flow rate of 5 L/h) or 10 h per day (ultrafiltration rate/dialysate flow rate of 4 L/h)] of hemofiltration or hemodialysis. Modeled doses were administered every 6 h, every 8 h, every 12 h, every 24 h, by extended infusion (4 h), by continuous infusion (24 h), or at the start (pre) and end (post) of SLED. To reach a PTA of ≥90%, piperacillin/tazobactam 4.5 g every 6 h was determined to be the optimal regimen to meet the efficacy target and with the lowest risk of toxicity based on trough concentrations. This dosing regimen met targets when 8-h hemodialysis was initiated at the same time as the first piperacillin-tazobactam dose or immediately post-SLED. Alternative regimens that met the PTA were 4.5 g every 6 h as an extended infusion (although a higher PTA was not obtained compared to conventional infusion time) or 16 g per day given as a continuous infusion. Piperacillin/tazobactam 16 g administered at the same time as the initiation of SLED was more likely to exceed the toxicity threshold than the other dosing regimens modelled [21].

A second study was a PK study of 34 adult ICU patients receiving piperacillin/tazobactam and 8-h SLED [22]. A Monte Carlo simulation was used to determine the optimal regimen. For a target of 50% of time above the MIC, 3 g of piperacillin infused over 30 min every 8 h was appropriate for pathogens with an MIC ≤16 mg/L to obtain a PTA of >90%. To achieve a PTA of >91% of 100% of time of the dosing interval above a MIC of ≤32 mg/L, 9 g of piperacillin was required to be given as a continuous infusion over 24 h. Toxicity was not assessed but no serious events, such as seizures, were reported [22].

Another PK study was performed on six critically ill patients with sepsis and anuric kidney failure requiring dialysis. Piperacillin-tazobactam (4/0.5 g) was administered over 30 min and a dose given 30 min prior to the initiation of SLED with a second dose given 12 h later in all patients except one who received a dose every 24 h [23]. Dialysis was administered for 6 h in all patients. Blood samples were taken after at least 2 days of SLED. During a 6-h SLED session 58% of piperacillin was cleared. The authors recommended for patients receiving 6-h SLED, the dosing of piperacillin should be at least 4 g every 12 h with at least a 2 g replacement dose post-SLED or 4 g every 8 h [23]. However, these results cannot be applied to durations of SLED other than 6 h.

### 3.2. Cephalosporins

#### 3.2.1. Ceftazidime

A MCS of free concentration of ceftazidime in SLED was performed which targeted >60% time greater than four times the MIC of *Pseudomonas aeruginosa* for the first 48 h of antibiotic therapy [21]. To obtain the PTA of ≥90%, ceftazidime 2 g every 12 h was determined to be the optimal regimen to meet the efficacy target and with the lowest risk of toxicity. This dosing regimen met targets when 8-h hemodialysis was initiated at the same time as the first ceftazidime dose or immediately post-dialysis. One gram of ceftazidime every 6 h or 3 g continuous infusion after 2 g loading dose also met PTA but had higher potential for toxicity if given at the same time as initiation of dialysis [21].

A second study was a PK study performed in an ICU setting in which serum samples were taken from 16 patients receiving 6 h of SLED [24]. Ceftazidime 1 g or 2 g was administered every 8–12 h over 30 min (dosing at physician discretion). Serum samples were drawn on three consecutive days of SLED to determine ceftazidime concentrations. A MCS was used to determine the PTA for the first 24 h of treatment. A dosing regimen of 1 g every 8 h achieved >95% PTA for 50% of time greater than the MIC. To achieve 100% of time above the MIC (as has been recommended for critically ill patients with severe infections), a dose of 2 g every 8 h was required for PTA of 99% and 2 g every 12 h for a PTA of 96% to cover strains with MICs up to 8 mg/L. The authors caution that these results cannot be extrapolated to SLED durations other than 6 h or if the SLED is interrupted because of the blood clotting in the filter. Ceftazidime is not recommended as monotherapy if the MIC of the pathogen is >8 mg/L. Toxicities were not reported [24].

#### 3.2.2. Cefepime

A MCS was performed of cefepime in SLED which targeted >60% time greater than four times the MIC of *Pseudomonas aeruginosa* for the first 48 h of antibiotic therapy [21]. To obtain a PTA of ≥90%, cefepime 1 g every 6 h after a loading dose of 2 g was determined to be the optimal regimen to meet the efficacy target with the lowest risk of toxicity. This dosing regimen met targets when 8-h hemodialysis was initiated at the same time as the first cefepime dose or immediately post-dialysis. Cefepime 2 g pre-dialysis plus 3 g post-dialysis also met the PTA [21].

#### 3.2.3. Ceftolozane/Tazobactam

A report described the pharmacokinetics of ceftolozane/tazobactam in one critically ill patient with a polymicrobial sternal wound osteomyelitis complicated by sepsis with a multidrug resistant *Pseudomonas aeruginosa* (MIC 4 mg/L) who underwent SLED [25]. A loading dose of ceftolozane/tazobactam of 500 mg/250 mg was administered intravenously over 90 min. A maintenance dose of 100/50 mg every 8 h was given on non-SLED days and 2 doses of 500/250 mg during and immediately after stopping SLED on dialysis days. Dialysis was run over 7.5 h and was initiated on day 3 of ceftolozane/tazobactam treatment. The concentrations of ceftolozane were at least two times higher than the MIC of the *Pseudomonas aeruginosa* throughout the entire sampling period. Clearance of ceftolozane and tazobactam increased during SLED vs. non-SLED periods (8.27 vs. 0.39 L/h and 8.02 vs. 0.77 L/h, respectively). The outcome for this patient was not reported. This study only included one patient who received one session of SLED; therefore, the results are difficult to generalize [25]. This report also does not include hospital- or ventilator-associated pneumonia patients where the recommended standard dosing is higher than for other indications.

### 3.3. Carbapenems

#### 3.3.1. Carbapenems

Monte Carlo simulations were used to determine initial dosing regimens for doripenem, imipenem, meropenem, and ertapenem in patients receiving SLED [26]. Simulations were done with 5000 virtual subjects to evaluate multiple dosing regimens and for four different SLED regimens/settings (8 or 10 h and SLED pre- vs. post-carbapenem dose). The PTA was calculated based on 40% of the fraction of time with free serum concentrations >4 times the MIC for the first 48 h. Optimal regimens were defined as a PTA in ≥90%. For *Pseudomonas aeruginosa* with an MIC = 2 mg/L the optimal dose of doripenem was 750 mg every 8 h, imipenem 1 g every 8 h or 750 mg every 6 h, and meropenem 1g every 12 h or 1 g pre- and 1 g post-SLED. For *Streptococcus pneumonia* with an MIC = 1 mg/L, ertapenem 500 mg followed by 500 mg post-SLED was optimal. Carbapenems with longer half-lives were more greatly impacted by the administration time of the antibiotic compared to SLED initiation than drugs with shorter half-lives. Toxicities were not studied but the minimum doses to achieve the PTA were recommended [26]. A limitation of these dosing regimens is the potential lack of efficacy if the organism has a higher MIC.

#### 3.3.2. Meropenem

A PK study was done on 19 critically ill septic patients receiving meropenem and SLED [27]. Treatment doses of 0.5 g, 1 g, or 2 g of meropenem were administered every 8 h at the discretion of the treating physician. Monte Carlo simulations were performed to determine the PTA for the first 24 h of meropenem treatment. The target was set as 40% of time greater than the MIC. The PTA was considered adequate if it was >95%. In patients with 0–100 mL/d of residual urine output the recommended dose was 0.5 g every 8 h. This dosing was also found to cover *Pseudomonas aeruginosa* (MIC ≤ 8 mg/L). If patients have a residual degree of diuresis of 300 mg/d, a dose of 1 g every 8 h is required to meet the specified target PTA. Higher doses were required if *Pseudomonas aeruginosa* had an MIC ≤ 16 mg/L. The authors concluded adequate meropenem dosing greatly varied depending on the residual level of urine output [27]. However, the average duration of SLED in this study was only 5 h and this may underestimate the dosing recommendations.

A second study included ten ICU patients with acute or chronic kidney failure requiring 8-h SLED [28]. Meropenem was given as 1g IV administered over 30 min every 12 h with at least two doses being given prior to the initiation of the study to ensure steady-state. Dialysis was initiated two to four hours post-meropenem dosing. The mean reduction of plasma meropenem concentration was 79.1 ± 7.3% and the mean half-life was 3.6 ± 0.8 h during SLED. Mean serum concentrations were reduced by 79.1% ± 7.3%. Meropenem was cleared more rapidly in the first four hours of SLED compared to the second four hours (66.5 ± 11.1% vs. 36.8 ± 18.3%). With an MIC_90_ = 2 mg/L for *Pseudomonas aeruginosa* as a reference point, the authors found the meropenem concentration was greater than the MIC_90_ for 100% of the dosing interval while on SLED [28]. However, for more resistant organisms with higher MICs, this dosing may be inadequate. A limitation of this study is that the blood flow rate (100–250 mL/min), dialysate flow rate (100–200 mL/min) and ultrafiltration rates (20–200 mL/h) were not standardized among patients which is known to affect drug clearance.

A PK study of 10 patients in a surgical ICU with acute kidney failure examined meropenem 1 g infused over 30 min given 6 h prior to SLED [29]. The fraction of meropenem removed by one dialysis session was 18%. The half-life was found to be 3.7 h while on SLED and 8.7 h while off dialysis [29]. No adverse effects were reported. The authors warned of a risk of underdosing in patients on SLED but did not provide a recommended dosing regimen for meropenem.

#### 3.3.3. Ertapenem

A PK study of ertapenem was done in six critically ill patients with acute anuric kidney failure undergoing 8-h SLED [30]. Ertapenem was administered as 1 g intravenously over 30 min. Plasma concentrations of ertapenem were greater than a MIC_90_ value of 2 mg/L for over 20 h after dosing. The clearance of ertapenem while on SLED was 49.5 mL/min, which was similar to ICU patients without kidney impairment (43.2 mL/min) or healthy volunteers (48.0 mL/min). The authors concluded patients treated with SLED likely require full dosing of ertapenem 1 g IV every 24 h [30]. This was only a single dose study so it is unclear if ertapenem accumulation is a concern over time in patients on SLED.

### 3.4. Colistin

A prospective PK single- and multiple-dose study of colistin and 8-h of SLED was performed in eight ICU patients [31]. Six million international units (MIU) of colistin methanesulfonate (CMS) were administered 8 h prior to the SLED session and then 3 MIU every 8 h. Multiple blood samples were taken on colistin day 1 and day 5 to determine colistin and CMS serum concentrations. Dialysis eliminated approximately half of the daily administered colistin dose. Despite a loading dose, therapeutic serum levels of colistin could not be achieved within the first 8 h of therapy in all patients. This is potentially due to the delayed conversion of CMS to active colistin in critically ill patients. A significant inverse correlation of body weight and peak concentrations of colistin were found, indicating that obese or volume-overloaded patients may require higher doses. Colistin was shown to accumulate after several days of therapy and could potentially lead to adverse effects; however, none were noted in this study. A limitation was that four of the eight patients died prior the data collection of colistin day 5 and two more died after day 9. Ultimately the authors suggested a loading dose of 6-9 MIU of CMS in patients >70 kg and/or with pathogens with high MIC values and a maintenance dose of 1.5-2 MIU every 8 h [31].

A case report was published of a patient with respiratory failure post lung transplant. The patient grew multidrug resistant *Klebsiella pneumoniae* in a lung sample [32]. The patient was given a loading dose of 6 MIU of CMS and a maintenance dose of 3 MIU of CMS every 8 h. The patient was started on SLED for acute kidney injury. After the loading dose, peak levels of colistin and CMS were 10.01 µg/mL and 24.76 µg/mL. After 9 days of the maintenance dose of colistin, no accumulation of colistin (peak level day 9: 8.96 µg/mL, trough level 2.13 µg/mL) nor CMS was seen. The clearance of colistin was 54 to 71 mL/min. The amount of colistin collected in the dialysate was 245 mg on day 1 and 191 mg on day 9. The patient died after 5 weeks in the ICU from a cerebral *Aspergillus* infection. The authors concluded that a dose of 3 MIU of CMS every 8 h is likely adequate for a patient undergoing daily dialysis for ~9 h per session and does not appear to lead to accumulation [32].

### 3.5. Fluoroquinolones

#### 3.5.1. Ciprofloxacin and Levofloxacin

A MCS modeled four different SLED regimens beginning at the time of or 14–16 h after fluoroquinolone administration [33]. Pharmacokinetic targets were AUC_24h_: MIC ratio of ≥125 for Gram-negative infections and ≥50 for Gram positive infections up to 72 h. The optimal dosing of ciprofloxacin to attain a PTA of 90% for a Gram-negative infection with *Pseudomonas aeruginosa* at the MIC of 1 mg/L was a loading dose of 1200 mg then 800 mg every 12 h. The optimal dosing of levofloxacin to attain 90% PTA for *Pseudomonas aeruginosa* with an MIC of 2 mg/L was a 2000 mg loading dose and 1000 mg every 24 h post-SLED. These recommendations exceed the maximum FDA-approved doses and the authors conclude that ciprofloxacin and levofloxacin cannot be recommended as empiric monotherapy for Gram-negative infections in patients receiving SLED due to suboptimal efficacy. Based on MCS, levofloxacin could only successfully attain PTA 90% for *Streptococcus pneumoniae* with an MIC of 1 mg/L. If a fluoroquinolone is being used in combination with another antimicrobial the recommended dosing is ciprofloxacin 400 mg IV every 8 h or levofloxacin with a loading dose of 750 mg IV and 750 mg post-SLED [33].

#### 3.5.2. Moxifloxacin and Levofloxacin

A study was performed of adult ICU patients with anuric acute kidney failure being treated with SLED [34]. Ten of these patients received moxifloxacin 400 mg IV infused over 60 min given 8 h prior to SLED. Five patients received levofloxacin at 250 mg or 500 mg IV infused over 60 min given 12 h prior to SLED. The clearance of moxifloxacin off dialysis was 15.7 L/h with a half-life of 12.3 h. While on SLED, the clearance of moxifloxacin increased between 2.0 to 3.1 L/h and the half-life was reduced to 6 h (range 3.9 to 11.0 h). The levofloxacin clearance off dialysis was 3.07 L/h and half-life was 34.5 h. Dialysis increased the clearance between 2.93 to 3.12 L/h and reduced the half-life to 10.3 h (range 10.0 to 10.6 h). Twenty to thirty percent of levofloxacin was removed by SLED. No adverse events attributable to the antibiotics were reported. The authors concluded that no dosage adjustment was required for moxifloxacin [34]. A dosing recommendation could not be made for levofloxacin but suggested it be administered post-SLED.

### 3.6. Vancomycin

A MCS was performed on thousands of virtual patients to determine the optimal dosing to achieve a vancomycin AUC_24h_: MIC ratio of ≥400 mg/L/h at 48 h for ≥90% of patients [35]. An AUC_24h_ of <700 mg/L/h was set to minimize the risk of toxicity. Four different daily SLED regimens were incorporated into the model. Nine different vancomycin dosing strategies were used. Dialysis lowered the vancomycin concentration by an average of 50% during the session. The optimal dosing was found to be an initial loading dose of 15–20 mg/kg given on initiation of SLED followed by a maintenance regimen of 15 mg/kg after each SLED session. If vancomycin is to be initiated and dialysis will not start for ≥12 h, the optimal dosing is 20 mg/kg initially, then 15 mg/kg post-SLED. The effluent rates and duration had an insignificant impact on the PTA of each dosing regimen [35]. However, many of the virtual patients could have developed potential toxicity as the AUC_24_ ≥700 mg/L/h.

A PK study examined vancomycin in eleven critically ill medical and surgical patients with acute kidney failure requiring SLED [36]. Dialysis ran continuously over 24 h and patients were given 15 mg/kg by actual body weight of vancomycin intravenously. Patients were re-dosed if the serum concentration was less than 20 mcg/mL at 24 h post-infusion. If the serum concentration 24 h post-infusion was 20–30 mcg/mL, another level was drawn at 40 h post-infusion. The mean half-life was calculated to be 43.1 h (range 18.8 to 96 h) and the clearance 24.3 mL/min (range 15–42 mL/min). The required dosing ranges to administer 20 mg/kg of vancomycin ranged from every 24 to 72 h. No vancomycin-related adverse effects were reported. Slow low efficiency dialysis was used as a continuous dialysis in this study, so it is difficult to extrapolate the results to shorter durations of intermittent SLED. The PK parameters showed great interpatient variability. The authors recommended an initial dose of vancomycin of 15 mg/kg of actual body weight and to check the vancomycin concentration 24-h post-dose [36]. However, newer vancomycin dosing guidelines recommend targeting AUC/MIC ratios rather than through levels when adjusting vancomycin dosing [37].

A third study examined population PK of vancomycin in eleven critically ill patients receiving SLED over 6–8 h [38]. Dialysis therapy did not correspond with the timing of vancomycin administration in all cases. Monte Carlo simulations were used to determine the PTA of an AUC_0-24_/MIC target of 400 and a target of AUC_0-24_>700 mg/L/h, (i.e., the breakpoint thought to be associated with increased risk of nephrotoxicity) [39]. A target of 90% PTA was set a priori. This group also reported high variability in the pharmacokinetics from patient to patient. The mean clearance of vancomycin on SLED was 3.47 L/h ± 1.99 L/h (57.8 mL/min ± 33.2 mL/min). A recommended dose to maximize the efficacy and minimize toxicity was vancomycin 25 mg/kg per day for patients receiving 12-h SLED to target organisms with an MIC of 1 mg/L [38]. The effects of >24 h of vancomycin therapy were not examined; therefore, it is difficult to develop a dosing regimen based on this study.

A PK study of ten patients in a surgical ICU with acute kidney failure examined vancomycin 1 g infused over 60 min given 12 h prior to SLED [29]. The fraction of vancomycin removed by one dialysis session was 26%. The half-life was found to be 11.2 h while on SLED and 37.3 h while off dialysis. No adverse effects were reported. The authors only recommended therapeutic drug monitoring to determine maintenance dosing [29].

### 3.7. Sulfamethoxazole/Trimethoprim (SMX/TMP)

A case was reported of a critically ill patient with *Pneumocystis jirovecii pneumonia* (PJP) treated with TMP 10 mg/kg/day and SMX 48 mg/kg/day in four divided doses [40]. Acute on chronic oliguric kidney injury occurred and SLED was initiated (average dialysis time 442 ± 101 min). The doses were increased to TMP 15 mg/kg/day and SMX 95 mg/kg/day. Peak drug concentrations in blood over three consecutive days of TMP and SMX were 7.51 mg/L and 80.80 mg/L, respectively, which are in the upper recommended range. The percentage of drug cleared during the dialysis session was 64% of TMP and 84% of SMX. The PJP resolved based on bronchoscopy results but the patient died in the ICU on day 35 due to a myocardial infarction [40].

### 3.8. Daptomycin

A case report was published on the PKs of daptomycin in a critically ill patient with acute infective endocarditis with negative blood cultures [41]. The patient developed septic shock and AKI. Daptomycin 6 mg/kg (i.e., 660 mg) based on actual body weight was administered over 30 min. Slow low efficiency dialysis was performed over 12 h and the dialysate collected to determine the quantity of daptomycin removed. The amount of daptomycin collected in the dialysate was 346 mg or 52% of the administered dose. This was a much higher clearance compared to IHD of 15%. The authors concluded the recommended dosing of every 48 h for IHD would be inadequate for SLED [41].

A single-dose PK study was performed in ten critically ill patients with anuric AKI being treated with 8-h SLED [42]. Patients received a single dose of daptomycin 6 mg/kg IV (based on actual body weight) infused over 30 min given 8 h prior to dialysis. It was found that half-life for daptomycin was comparable in ED patients while on dialysis to healthy controls (8.0 ± 1.8 h vs. 7.8 ± 1.0 h). The mean fraction of the drug removed by one ED session was 23.3%. The authors recommended a dose of 6 mg/kg per day of daptomycin to avoid underdosing and concluded the IHD dosing recommendation of 6 mg/kg every 48 h would be inadequate for patients receiving SLED [42]. Since this was a single dose study, it is unclear if daptomycin will accumulate when multiple doses are administered.

### 3.9. Linezolid

A 24-h PK study involved 15 adult critically ill surgical patients with sepsis being treated with linezolid [43]. Ten of these patients were receiving SLED for acute anuric kidney failure. Linezolid 600 mg intravenously was infused twice daily over 1 h. The average dialysis time was 19.5 h (range 12–24 h). Blood samples were drawn multiple times over 4 days. Linezolid was administered at time zero and 12 h. Slow low efficiency dialysis increased the clearance of linezolid by 23% and the trough concentrations were often below the susceptibility breakpoint and only transiently above the MIC. Outcomes for these patients were not reported and no adverse events related to linezolid were identified. Generalizability of the results to smaller patients is unclear as the mean BMI of these patients was over 30 kg/m^2^. No recommendations for higher dosing were provided and it was suggested that clinicians should attempt to utilize therapeutic drug monitoring to optimize drug levels [43].

A case report was published of a man with biliary tract sepsis, oliguric AKI, and chronic liver disease admitted to ICU receiving SLED 6–8 h per day [44]. He was initiated on multiple antibiotics including linezolid 600 mg IV twice daily. Cultures from blood and biliary fluid grew vancomycin-resistant *Enterococcus faecium.* Samples from blood, bile, and peritoneal fluid were drawn before and after each dose of linezolid and at the start and end of each SLED session. It was determined that the serum and bile AUC/MIC ratios were adequate. The patient received surgery seven days after hospital admission and post-operative blood cultures were negative but the patient expired four days later due to intractable arrhythmias [44]. These results may not be applicable to patients on SLED without hepatic impairment, as 50–70% of linezolid is metabolized by the liver and this likely contributed to this patient having adequate pharmacokinetic parameters.

A single-dose PK study was performed in five critically ill patients with oliguric acute kidney failure receiving 8-h SLED [45]. These patients received 600 mg IV of linezolid infused over 60 min prior to dialysis. It was found that 33.9% ± 13% of the drug was removed during an 8-h SLED session. In three of five patients, serum linezolid levels were <4 g/L at the end of SLED and could potentially lead to reduced efficacy. No dosing recommendations were made as this was a single dose study in a small sample of patients [45].

### 3.10. Antifungals

#### 3.10.1. Anidulafungin

A case study of a critically ill man with *Candida albicans* cholecystitis, sepsis, and AKI examined the pharmacokinetics of anidulafungin during SLED [46]. Anidulafungin 200 mg was administered over 30 min. A sample of the dialysate was collected at the end of the 8-h dialysis period. The amount of anidulafungin in the dialysate was undetectable. Pharmacokinetic data was comparable to healthy adults. Therefore, it was concluded based on this single case that anidulafungin does not require dose adjustment for SLED [46].

#### 3.10.2. Fluconazole

A MCS was performed to determine the optimal regimen of fluconazole dosing in SLED of 8–10 h [47]. The optimal dosing was based on probability of attaining a mean 24-h AUC to MIC ratio of ≥100 during the initial 48 h of fluconazole therapy. Multiple dosing regimens were simulated in 5000 subjects. At a breakpoint of MIC = 2 mg/L for *C. albicans*, 93–96% of the simulated population reached the target with a loading dose of fluconazole 800 mg IV then 400 mg IV twice daily. This could be administered either every 12 h or pre- and post-SLED. The authors did not anticipate toxicity with these doses as the simulated maximum concentrations ranged between 35–60 mg/L. The primary toxicity of hepatic injury is usually observed at concentrations exceeding 70 mg/L; however, the authors recommend monitoring for liver toxicity. This recommended dosing may be inadequate for patients growing non-*C. albicans* species or with higher MICs. The probability of target attainment was also found to be lower in patients with higher body weights [47].

#### 3.10.3. Voriconazole

A concern with voriconazole parenteral formulation in kidney impairment is the solubilizing agent sulphobutylether-β-cyclodextrin (SBECD). It is cleared renally with a warning against use in acute or end-stage kidney insufficiency and hemodialysis in the product monograph. The accumulation of SBECD was examined in four critically ill patients with anuric acute kidney failure receiving 8-h SLED [48]. The patients were given voriconazole 4 mg/kg IV every 12 h. A clear accumulation of SBECD was seen on day 5 based on higher peak and trough levels, an increased AUC and elimination half-life. No accumulation of voriconazole was seen. Animal studies have shown necrosis and obstruction of kidney tubes and liver toxicity due to SBECD accumulation. However, no toxicities were reported for the four patients in this report [48].

## 4. Conclusions

Drug dosing in extended modes of dialysis used in critically ill patients is highly challenging. There are substantial changes in the pharmacokinetics and pharmacodynamics in this population. A paucity of studies has been published in this setting and the results can be difficult to extrapolate because of their small sample size, variability in durations of dialysis, blood and dialysate flow, and timing of drug administration in relationship to dialysis. Despite an extensive literature review, some antimicrobials such as aminoglycosides and amphotericin B have no published information on SLED dosing. From the studies published, it appears dosing recommendations for other modes of dialysis such as IHD or CRRT are highly inadequate in SLED and will potentially lead to antibiotic failure. No literature is available on dosing of non-antimicrobials in extended modes of dialysis and remains an area requiring further research as these modes of dialysis are increasing in popularity.

## Figures and Tables

**Table 1 pharmacy-08-00033-t001:** Dialysis modalities for the critically ill patient.

Modality	Acronym	Description	Blood Flow Rate (mL/min)	Dialysate Flow Rate (mL/min)
Continuous Renal Replacement Therapy *	CRRT	Generic term to describe dialysis over 24 h	10–180	0–45 (0–2.5 L/h) ^
Intermittent Hemodialysis	IHD	Conventional intermittent dialysis over 4 h, 3 times per week	250–400	200–350
Sustained Low efficiency dialysis **	SLED	Dialysis over 6–12 h	200–300	200–350
Slow extended daily dialysis **	SLEDD	Dialysis over 6–12 h	same as above	same as above

* CRRT encompasses different dialysis and hemofiltration modalities. ^ dialysate flow rate reported as L/h. ** Prolonged intermittent renal replacement therapy (PIRRT) and extended daily dialysis (EDD) are also other terms used for a slow extended dialysis with similar parameters to SLED/SLEDD.

**Table 2 pharmacy-08-00033-t002:** Summary of published trials examining drug dosing in extended modes of dialysis of Penicillins.

Drug	Study Design	Dose	Single vs. Multiple Doses	# of Subjects	Mode of Dialysis	Dialysis Duration (min) (mean ± SD)	Blood Flow (mL/min) (mean ± SD)	Dialysate Flow (mL/min) (mean ± SD)	Machine and Filter Information	Dosing Recommendation as per Study
Penicillin G [18]	PK	3 MU q6h with a dose given within the first hour of PIRRT and a dose within one hour of stopping PIRRT	Multiple (48 h)	2	PIRRT	510 or 570	200	200	Fresenius 5008; Ultraflux AV600S, SA 1.4 m^2^	1800 mg (3 MU) q6h with doses given within the first hour of PIRRT and within the first hour of stopping PIRRT
Ampicillin/sulbactam [19]	PK	3 g (ampicillin 2 g/sulbactam 1 g) over 30 min given 4 h prior to EDD	Single	1	EDD	450	180	180	GENIUS, PS high-flux (F60S), SA 1.3 m^2^	No specific recommendation but suggested IHD dosing is inadequate
Ampicillin/sulbactam [20]	PK	3 g (ampicillin 2 g/sulbactam 1 g) over 30 min given 3 h prior to ED	Single and multiple (4 d in 3 subjects)	12	ED	442 ± 77	162 ± 6	162 ± 6	GENIUS, PS high flux (F60S); SA 1.3 m^2^	At least 2 g/1 g ampicillin/sulbactam BID with one dose given post ED
Piperacillin/tazobactam [21]	MCS	Multiple dosing models with dose given just prior or post-SLED	Multiple (48 h)	5000 virtual	PIRRT	480 or 600	300	66.7 or 83.3	N/A	4.5 g q6h if PIRRT initiated at the same time as the first piperacillin-tazobactam dose or immediately post-PIRRT
Piperacillin/tazobactam [22]	PK & MCS	3 g of piperacillin q8h over 30 min; for MCS multiple dosing regimens modeled	Multiple	PK = 34 MCS = 5000 virtual	SLED	480	200	300	Gambro Artis; high flux F40S PS, SA 0.7 m^2^	3.375 g of piperacillin/tazobactam administered over 30 min q8h for pathogens with MIC <16 mg/L; for life-threatening infections this dose should be given as a continuous infusion
Piperacillin/tazobactam [23]	PK	Piperacillin 4 g/tazobactam 0.5 g q12h given over 30 min given 30 min prior to SLED-f	Multiple	6	SLED-f	360	200	200	4008S; A600S PS filter, SA 1.4 m^2^	At least piperacillin 4 g q12h with 2 g post-SLED-f or 4 g q8h

BID = twice daily; d = days; ED = extended dialysis; EDD = extended daily dialysis; g = grams; h = hours; IV = intravenous; MCS = Monte Carlo simulation; MD = maintenance dose; mg = milligrams; min = minutes; MU = million units; N/A = not available; PIRRT = prolonged intermittent renal replacement therapy; PK = pharmacokinetic; PS = polysulfone; q = every; SA = surface area; SLED = slow low efficiency dialysis; SLED-f = slow low efficiency diafiltration.

**Table 3 pharmacy-08-00033-t003:** Summary of published trials examining drug dosing in extended modes of dialysis of Cephalosporins.

Drug	Study Design	Dose	Single vs. Multiple Doses	# of Subjects	Mode of Dialysis	Dialysis Duration (min) (mean ± SD)	Blood Flow (mL/min) (mean ± SD)	Dialysate Flow (mL/min) (mean ± SD)	Machine and Filter Information	Dosing Recommendation as per Study
Ceftazidime [21]	MCS	Multiple dosing models with dose given just prior or post-SLED	Multiple (48 h)	5000 virtual	PIRRT	480 or 600	300	66.7 or 83.3	N/A	2 g q12h if PIRRT initiated at same time as ceftazidime dose or immediately post-PIRRT
Ceftazidime [24]	PK & MCS	1 g or 2 g q8h or q12h administered over 30 min	Multiple (24 h)	PK = 16 MCS = 1000 virtual	SLED	299 ± 68.4	264 ± 40.4	264 ± 40.4	GENIUS, Fresenius FX 60 filter, SA 1.4 m^2^	2 g q8h or 2 g q12h (for MICs ≤8 mg/L); ceftazidime not recommended for monotherapy if MIC >8 mg/L
Cefepime [21]	MCS	Multiple dosing models with dose given just prior or post-SLED	Multiple (48 h)	5000 virtual	PIRRT	480 or 600	300	66.7 or 83.3	N/A	LD of 2 g then 1 g q6h when PIRRT is initiated at same time as first cefepime dose or immediately post-PIRRT
Ceftolozane/tazobactam [25]	PK	500/250 mg administered over 90 min, 100/50 mg q8h for non-PIRRT days and 500/250 mg during and post-PIIRT on dialysis days	Multiple	1	PIRRT	450	200	200	Fresenius 5008; Ultraflux AV600S, SA 1.4 m^2^	500 mg/250 mg during and after PIRRT, 100/50 mg q8h during non-PIRRT periods for *P. aeruginosa* with MIC ≤ 4 mg/L

d = days; g = grams; h = hours; IV = intravenous; kg = kilogram; L = liter; LD = loading dose; MCS = Monte Carlo Simulation; MD = maintenance dose; MIC = minimum inhibitory concentration; mg = milligrams; min = minutes; mL = milliliters; MU = million units; N/A = not available; PIRRT = prolonged intermittent renal replacement therapy; PK = pharmacokinetic; PS = polysulfone; q = every; SA = surface area; SLED = Slow Low Efficiency Dialysis.

**Table 4 pharmacy-08-00033-t004:** Summary of published trials examining drug dosing in extended modes of dialysis of Carbapenems.

Drug	Study Design	Dose	Single vs. Multiple Doses	# of Subjects	Mode of Dialysis	Dialysis Duration (min) (mean ± SD)	Blood Flow (mL/min) (mean ± SD)	Dialysate Flow (mL/min) (mean ± SD)	Machine and Filter Information	Dosing Recommendation as per Study
Meropenem [26]	MCS	Multiple dosing regimens modeled given pre- or post-PIRRT	Multiple (48 h)	5000 virtual	PIRRT	480 or 600	300	66.7 or 83.3	N/A	1 g q12h or 1 g pre- and post- PIRRT for *P. aeruginosa* with MIC of 2 mg/L
Meropenem [27]	PK & MCS	0.5 g, 1 g, or 2 g administered over 30 min given q8h; SLED started within 3 h of dose	Multiple (24 h)	PK = 19 MCS = 1000 virtual	SLED	315 [range 275–354]	250 [range 208–278]	250 [range 208–278]	GENIUS, Fresenius FX 60 filter, SA 1.4 m^2^	Dosing requirements dependent on the degree of residual urine output (RUO) −500 mg q8h if RUO was 0–100 mL/d; −1 g q8h if RUO >300 mL/d
Meropenem [28]	PK	1 g administered over 30 min q12h given at 2–4 h prior to SLED	Single	10	SLED	480	160 ± 45.9	170 ± 42.2	Fresenius 2000K with AV 400 PS dialyzer, SA 0.7 m^2^	1 g q12h for *P. aeruginosa* with MIC = 2 mcg/mL
Meropenem [29]	PK	1 g IV over 30 min given 6 h prior to EDD	Single	10	EDD	480 ± 6	160	160	GENIUS, PS high-flux (F60S); SA 1.3 m^2^	500 mg to 1000 mg q8h but should be tailored to severity of illness and MIC of organism
Ertapenem [26]	MCS	Multiple dosing regimens modeled, given pre- or post-PIRRT	Multiple (48 h)	5000 virtual	PIRRT	480 or 600	300	66.7 or 83.3	N/A	500 mg followed by 500 mg post-PIRRT for *Streptococcus pneumoniae* with MIC of 1 mg/L
Ertapenem [30]	PK	1 g IV administered over 30 min	Single	6	EDD	480	160	160	GENIUS, PS high flux (F60S), SA 1.3 m^2^	1 g per day
Doripenem [26]	MCS	Multiple dosing regimens modeled, given pre- or post-PIRRT	Multiple (48 h)	5000 virtual	PIRRT	480 or 600	300	66.7 or 83.3	N/A	750 mg q8h for *P. aeruginosa* with MIC of 2 mg/L
Imipenem [26]	MCS	Multiple dosing regimens modeled, given pre- or post-PIRRT	Multiple (48 h)	5000 virtual	PIRRT	480 or 600	300	66.7 or 83.3	N/A	1g q8h or 750 q6h for *P. aeruginosa* with MIC of 2 mg/L

BID = twice daily; d = days; EDD = extended daily dialysis; g = grams; h = hours; IV = intravenous; kg = kilogram; L = liter; LD = loading dose; MCS = Monte Carlo simulation; MD = maintenance dose; MIC = minimum inhibitory concentration; mg = milligrams; min = minutes; mL = milliliters; N/A = not available; PIRRT = prolonged intermittent renal replacement therapy; PK = pharmacokinetic; PS = polysulfone; q = every; SA = surface area; SLED = slow low efficiency dialysis.

**Table 5 pharmacy-08-00033-t005:** Summary of published trials examining drug dosing in extended modes of dialysis of Fluoroquinolones.

Drug	Study Design	Dose	Single vs. Multiple Doses	# of Subjects	Mode of Dialysis	Dialysis Duration (min) (mean ± SD)	Blood Flow (mL/min) (mean ± SD)	Dialysate Flow (mL/min) (mean ± SD)	Machine and Filter Information	Dosing Recommendation as per Study
Ciprofloxacin [33]	MCS	Multiple regimens modeled with doses given pre-PIRRT and post-PIRRT	Multiple (72 h)	5000 virtual	PIRRT	480 or 600	300	66.7 or 83.3	N/A	Required doses exceeded FDA max doses; not recommended as empiric monotherapy If used as combination therapy LD of 400 mg then 400 mg q8h
Levofloxacin [33]	MCS	Multiple dosing regimens modeled with doses given pre-PIRRT and post-PIRRT	Multiple (72 h)	5000 virtual	PIRRT	480 or 600	300	66.7 or 83.3	N/A	Required doses exceeded FDA max doses; not recommended as empiric monotherapy If used as combination therapy LD of 750 mg then 750 mg q24 h post-PIRRT
Levofloxacin [34]	PK	250 mg or 500 mg IV administered over 60 min given 12 h prior to EDD	Single	5	EDD	481 ± 9	160 ± 4	160 ± 4	GENIUS, PS high-flux dialyzer (F60S), SA 1.3 m^2^	No specific recommendation but should be administered post-EDD
Moxifloxacin [34]	PK	400 mg IV administered over 60 min given 8 h prior to EDD	Single	10	EDD	481 ± 9	160 ± 4	160 ± 4	GENIUS, PS high-flux dialyzer (F60S), SA 1.3 m^2^	400 mg IV once daily post-EDD

BID = twice daily; d = days; ED = extended dialysis; EDD = extended daily dialysis; g = grams; h = hours; IV = intravenous; kg = kilogram; L = liter; LD = loading dose; MCS = Monte Carlo simulation; MD = maintenance dose; MIC = minimum inhibitory concentration; mg = milligrams; min = minutes; mL = milliliters; N/A = not available; PIRRT = prolonged intermittent renal replacement therapy; PK = pharmacokinetic; PS = polysulfone; q = every; SA = surface area.

**Table 6 pharmacy-08-00033-t006:** Summary of published trials examining drug dosing in extended modes of dialysis of Vancomycin.

Drug	Study Design	Dose	Single vs. Multiple Doses	# of Subjects	Mode of Dialysis	Dialysis Duration (min) (mean ± SD)	Blood Flow (mL/min) (mean ± SD)	Dialysate Flow (mL/min) (mean ± SD)	Machine & Filter Information	Dosing Recommendation as per Study
Vancomycin [35]	MCS	Nine regimens modeled with doses given immediate pre-PIRRT or post-PIRRT	Multiple (48 h)	5000 virtual	PIRRT	480 or 600	300	83.3 or 66.7	N/A	LD of 15–20 mg/kg with initiation of PIRRT; MD of 15 mg/kg after each PIRRT session plus TDM If vancomycin is to be initiated and PIRRT will not start for ≥12 h LD 20 mg/kg; MD 15 mg/kg post-PIRRT plus TDM
Vancomycin [36]	PK	15 mg/kg by actual body weight	Multiple	11	SLED	continuous	200	100	Fresenius 2008H; PS low flux (F4 or F5); SA 1.2 m^2^	Initial dose of 15 mg/kg of actual body weight with levels drawn 24 h following the initial dose
Vancomycin [38]	PK & MCS	Dose given 12 h prior to PIRRT; Multiple dosing regimens modeled for MCS	Single or multiple	PK = 11 MCS =1000 virtual	PIRRT	360 or 480	300	300	Fresenius 4008S, AV600S, SA 1.4 m^2^ or Genius, PS high-flux dialyzer (F60S), SA 1.3 m^2^	25 mg/kg/d in 12 h PIRRT with TDM
Vancomycin [29]	PK	1 g IV over 60 min given 12 h prior to EDD	Single	10	EDD	480 ± 6	160	160	GENIUS, PS high-flux (F60S); SA 1.3 m^2^	Initial dose of 20–25 mg/g then TDM

BID = twice daily; d = days; ED = extended dialysis; EDD = extended daily dialysis; g = grams; h = hours; IV = intravenous; kg = kilogram; L = liter; LD = loading dose; MCS = Monte Carlo simulation; MD = maintenance dose; MIC = minimum inhibitory concentration; mg = milligrams; min = minutes; mL = milliliters; N/A = not available; PIRRT = prolonged intermittent renal replacement therapy; PK = pharmacokinetic; PS = polysulfone; q = every; SA = surface area; SLED = slow low efficiency dialysis; TDM = therapeutic drug monitoring.

**Table 7 pharmacy-08-00033-t007:** Summary of published trials examining drug dosing in extended modes of dialysis of other antibiotics.

Drug	Study Design	Dose	Single vs. Multiple Doses	# of Subjects	Mode of Dialysis	Dialysis Duration (min) (mean ± SD)	Blood Flow (mL/min) (mean ± SD)	Dialysate Flow (mL/min) (mean ± SD)	Machine and Filter Information	Dosing Recommendation as per Study
Colistin [31]	PK	6 MU CMS given 8 h prior to PIRRT then 3 MU q8h given over 30 min	Single & Multiple	8	PIRRT	480	200	200	GENIUS; PS high-flux; SA 1.3 m^2^	LD of 6–9 MU CMS in patients >70 kg and/or with pathogens with high MIC values, MD of 1.5–2 MU every 8 h Obese or volume-overloaded patients may require higher doses
Colistin [32]	PK	6 MU CMS then 3 MU q8h	Single and Multiple (9 d)	1	ED	552	191	121	High-flux, SA 1.3 m^2^	3 MU CMS every 8 h
Sulfamethoxazole/ trimethoprim [40]	PK	SMX 95 mg/kg/d and TMP 15 mg/kg/d	Multiple	1	EDD	442 ± 101	170 ± 41	170 ± 41	GENIUS, PS high-flux dialyzer (F60S), SA 1.3 m^2^	No specific recommendations but dose reduction from standard may lead to underdosing
Daptomycin [41]	PK	6 mg/kg actual body weight over 30 min	Single	1	EDD	720	200	100	GENIUS, PS high-flux dialyzer (F60S), SA 1.3 m^2^	No specific recommendation but IHD dosing is likely inadequate
Daptomycin [42]	PK	6 mg/kg administered over 30 min given 8 h prior to ED	Single	10	ED	456 ± 13	166 ± 5	166 ± 5	GENIUS, PS high flux (F60S), SA 1.3 m^2^	6 mg/kg per day with ED starting within 8 h of dose
Linezolid [43]	PK	600 mg q12h infused over 60 min	Multiple (24h)	10	ED	1170 [range 720–1440]	110–150	N/A	GENIUS, PS high-flux dialyzer (F60S), SA 1.3 m^2^	No specific recommendation but higher doses may be required except in patients with concomitant liver failure
Linezolid [44]	PK	600 mg IV twice daily	Multiple	1	SLED	360–480	200	300	PS Fresenius F8 HPS filter, SA 1.6 m^2^	No specific recommendation; higher doses may not be required in patient with liver failure
Linezolid [45]	PK	600 mg IV over 60 min given prior to SLED	Single	5	SLED	480–540	200	100	PS low-flux (F7HPS), SA 1.6 m^2^ or BLS 514G, SA 1.4 m^2^	Give dose at end of SLED session

BID = twice daily; CMS = colistin methanesulfonate; d = days; ED = extended dialysis; EDD = extended daily dialysis; g = grams; h = hours; IV = intravenous; kg = kilogram; L = liter; LD = loading dose; m = meter; MCS = Monte Carlo simulation; MD = maintenance dose; MIC = minimum inhibitory concentration; mg = milligrams; min = minutes; mL = milliliters; MU = million units; N/A = not available; PIRRT = prolonged intermittent renal replacement therapy; PK = pharmacokinetic; PS = polysulfone; q = every; SA = surface area; SLED = slow low efficiency dialysis; SLED-f = slow low efficiency diafiltration; SMX= sulfamethoxazole; TDM = therapeutic drug monitoring; TMP = trimethoprim.

**Table 8 pharmacy-08-00033-t008:** Summary of published trials examining drug dosing in extended modes of dialysis of antifungals.

Drug	Study Design	Dose	Single vs. Multiple Doses	# of Subjects	Mode of Dialysis	Dialysis Duration (min) (mean ± SD)	Blood Flow (mL/min) (mean ± SD)	Dialysate Flow (mL/min) (mean ± SD)	Machine and Filter Information	Dosing Recommendation as per Study
Anidulafungin [46]	PK	200 mg IV administered over 30 min	Single	1	EDD	480	180	180	GENIUS, PS high-flux dialyzer (F60S), SA 1.3 m^2^	No dose adjustment required
Fluconazole [47]	MCS	Various dose regimens with the dose administered pre- or post-PIRRT	Multiple (48 h)	5000 virtual	PIRRT	480 or 600	300	N/A	N/A	LD of 800 mg then 400 mg twice daily
Voriconzole [48]	PK	4 mg/kg IV twice daily	Multiple (5 d)	4	EDD	480	180	180	GENIUS, PS high-flux dialyzer (F60S), SA 1.3 m^2^	Voriconazole IV cannot be recommended as SBECD accumulation was substantial

d = days; EDD = extended daily dialysis; g = grams; h = hours; IV = intravenous; kg = kilogram; L = liter; m = metre; MCS = Monte Carlo simulation; MIC = minimum inhibitory concentration; mg = milligrams; min = minutes; mL = milliliters; N/A = not available; PIRRT = prolonged intermittent renal replacement therapy; PK = pharmacokinetic; PS = polysulfone; q = every; SA = surface area; SBECD = sulphobutylether-β-cyclodextrin.

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
