# Peer review of "Principles of Drug Dosing in Sustained Low Efficiency Dialysis (SLED) and Review of Antimicrobial Dosing Literature"

_pharmacy, 2020, doi:10.3390/pharmacy8010033_

Round 1
Reviewer 1 Report
Please see the attachment.

Reviewer 2 Report
Brown et al. reported a literature review of antimicrobial dosing in sustained low efficiency dialysis. Even though this paper is adding a value to the current literature, it is missing numerous important data. For example, aminoglycosides, imipenem, amphotericin B, and new vancomycin consensus guideline are not mentioned. Authors should think about the structural changes for better flow of the manuscript. Paragraphs are “choppy” in the Principles of Drug Dosing in SLED and PK and PD principles during SLED sections. More specific comments are following:
Principles of Drug Dosing in SLED:
Line 29: Do you need both “greater” and “higher”?
Line 36: Please do not start the sentence with abbreviation
Line 40: Merge this paragraph with the previous one.
Line 44: “Less efficient” does not mean slower flow rates. Drug removal efficiency is: CVVHDF>CVVHD>CVVH>PIRRT>/=IHD (Hoff et al. Annals of Pharmacotherapy 2019)
Line 45: Please state dialyzers are also relatively inexpensive compared to…?
Line 49: Did you mean “hypophosphatemia” is developed from SLED itself? Or from critical illness? Please be clear.
Line 51: The reference should be 13, not 7. This sentence needs to be more specific. When you say “less leaky”, did you mean the hemodiafilter has a smaller surface area?
PK and PD principles:
Line 61 and 62: These paragraphs can be merged
Line 65-66: Oral medication are rarely influenced by the use of renal replacement therapy. Also, it is very short. Is this necessary to be included in this manuscript?
Line 70: …fluid overload secondary to excessive fluid administration “with no clearance function”
Line 80: “CKD patients” should be changed to “critically ill patients”. Also, add critically ill patients tend to be hypo-albuminuric (~3g/dL).
Line 84: Needs reference
Line 87-88: Is it necessary to define what drug clearance is for our audience?
Line 101: Wouldn’t Scoville and Mueller’s AJKD paper be better than ref. 15 here?
Line 103: Is continuous infusion the most appropriate drug administration technique?
Line 109: This should be part of the previous sentence (Line 107-108). Moreover, it may not be necessary to define what pharmacodynamics in our audience.
Line 115: vancomycin is not time-dependent. It has both time- and concentration-dependent anti-bactericidal and the PD target is stated as AUC/MIC, not T>MIC.
Line 116: There is a typo for “activity”
Line 118: “Often given more frequently at lower doses” does not support the PD target of 4XMIC nor 100% T>MIC for time-dependent antibiotics.
Table 1:
Please merge hybrid RRT since they are the same except for the names. I would provide the range for the blood and dialysate flow rate for hybrid RRT. Lastly, match the font of this table with the rest of the manuscript. Table 2 has the same font as the rest of the manuscript.
Published studies of …modes of dialysis section:
Line 129: Introduced abbreviation of pharmacokinetics for the first time. I would abbreviate PK from the introduction and use it.
Line 131: needs a reference after “minimizing toxicity”
Line 133: needs a reference
Line 136: …90% “probability of target attainment”…
Line 137: readers will need more information regarding the concept of 4XMIC and why it is beneficial.
Line 144-154: I am unsure if the Penicillin G case report is adding any value to this article. Maybe the table 2 is enough.
Line 162: implication of “~30% of the dosing interval” will need to be clearly stated for the readers. The first paragraph article of amp/sul emphasizes the underdosing in the patients. This should be clearly interpreted and stated in this paragraph.
Line 169-170; 192-193; 203-205: Not pertinent thus, may not need these information.
Line 174: “No outcome…patients” should be deleted. Not pertinent and it does not add any value to the literature.
Line 189: were there any toxicity concerns for different drug administration strategies?
In general for each drug paragraphs:
Be consistent with “SLED” terminology even though the primary literature from each article may use different terminology such as ED or SLED-f. It might be confusing for the readers. Please be consistent with the spacing between sentences throughout the manuscripts. Some sentences have one space but, some might have two spaces. Please only include pertinent information. For example, it is not necessary for audience to know how drug concentrations were measured (HPLC vs. LC-MS) or blood collection unless there is important implication regarding this. The purpose of this review is to guide the clinicians what to consider when they are drug dosing in patients receiving SLED. Usually, you do not start the sentences with abbreviations. No need to mention about outcome at this point especially there were only two patients or none. Please include the new vancomycin guideline and the implication of it (https://www.ashp.org/-/media/assets/policy-guidelines/docs/draft-guidelines/draft-guidelines-ASHP-IDSA-PIDS-SIDP-therapeutic-vancomycin.ashx)
Reference list:
Correct the duplicated numbers in the reference list from #5.Author Response
Please see attachment

Reviewer 3 Report
SLED is used as a renal replacement in critically ill patients. The appropriate dosing of medications for patients undergoing SLED is not well studied.
This review summarizes published trials of antimicrobial dose adjustment in SLED and discusses pharmacokinetics accordingly. This review is well written. However, some minor topics need to be discussed.
1. the changes of pharmacokinetic parameters are not well written. some more evidence-based references need to be included.
2. although this review focuses on human studies, some mechanisms underlying AKI/CKD need to be discussed.
For example, lots of animal studies tried to explain the effect of AKI/CKD on drug transporters and metabolism. Authors have to include a small part specifically discuss it.
Round 2
Reviewer 1 Report
Overall the manuscript is better, but the background stills need clean-up.
I am also concerned as when I checked the references it appears the verbiage used by the authors is very similar to the references cited such as the following example. Lines 39-48.
Also reference #7 for this manuscript and Reference #7 in the Mushatt DM paper (ref #13) are the same paper.
Berbece AN, Richardson RM. Sustained low-efficiency dialysis in the ICU: cost, anticoagulation, and solute removal, Kidney Int, 2006, vol.70(pg.963-8)
As with continuous therapies, hemodynamic stability in SLED is better than in intermittent hemodialysis because of decreased ultrafiltration rates and runs for a longer time (Table 1). Conventional dialysis machines are used; thus, no additional equipment is needed. [7]. The dialyzers are also - inexpensive. A standard dialysate concentrate is used, rather than specialized dialysate and ultrafiltrate replacement solutions [7]. Furthermore, anticoagulation is not generally necessary for SLED [7]. Overall, the costs are considerably less than those for CRRT [8]. There are some disadvantages to SLED such as unfamiliarity with the modality, and hypophosphatemia [7]. Finally, one other disadvantage with SLED is that there is less removal of “middle molecules” in the size range 1000–10,000 Daltons compared to CRRT as the dialyzer membranes with SLED are generally less –permeable [13].
Reference 13 Mushatt DM, Mihm LB, Dreisbach AW, et al. Antibiotic dosing in slow extended daily dialysis. Clin Infect Dis. 2009;49:433- 437.
As with continuous therapies, hemodynamic stability in SLEDD is better than in intermittent hemodialysis because it is still less efficient and runs for a longer time. Conventional dialysis machines are used; thus, no additional equipment is needed, and the machine can be used for intermittent hemodialysis. The dialyzers are also relatively inexpensive. Routine dialysate concentrate is used, rather than specialized dialysate and ultrafiltrate replacement solutions. Overall, the operating expenses are considerably less than those for CVVH [6]. Furthermore, anticoagulation is not generally necessary for SLEDD [7]. There was less removal of “middle molecules” in the size range 1000–10,000 Daltons than with CVVH, because the membranes are generally less leaky, and this may be a disadvantage
Throughout the manuscript the writing style uses “of” more than necessary for instance on line 99 “the clearance of the metabolites of the parent drug” the writing could easily be modified to a more succinct style.
The following system does not appear to match what your reference states. Please correct.
Lines 36-39. Some differences between SLED and 37 other traditional continuous therapies include: clearances for small molecules are generally higher 38 per hour than they are in CRRT; remembering, however, that SLED is generally used for only 6–12 h 39 per day and thus overall there is less clearance [8]
Copied from the published reference.
“All systems offer the advantages of flexible timing of treatment and reduced costs; their ease of handling means that SLED is readily accepted by ICU staff. Prospective controlled studies have shown that SLED clears small solutes with an efficacy comparable to that of intermittent hemodialysis and continuous venovenous hemofiltration (even when the latter employs high rates of fluid substitution).” Fliser D, Kielstein JT. Technology insight: treatment of renal failure in the intensive care unit with extended dialysis. Nat Clin Pract Nephrol 2006; 2:32–9.
Line 97 remove the word both from both human studies.
Line 500. Should a Heading of Conclusion be added here?
References. Authors names misspelled.
Villay MA, Churchill MD, Mueller BA. Clinical review: Drug metabolism and nonrenal clearance in 578 acute kidney injury. Crit Care. 2008; 12(6): 235-244.
Authors misspelled. Should be Vilay AM, Churchwell MD, Mueller BA
References Formatting is not consistent.
For Example: Some authors the Initials are B.A or J.T. but other are BA or JT.
Example. Pieter Evenepoel, Bjorn K.I.Meijers, Bert R.M.Bammens, KristinVerbeke. Uremic toxins originating 568 from colonic microbialmetabolism. Kidney International(2009)76(Suppl 114), S12–S19.
Author Response
Thank you to Reviewer #1 for comments and suggestions. We have made all the changes to the manuscript. We have addressed all the issues below and responded to the comments below:
Reviewer #1: Overall the manuscript is better, but the background stills need clean-up.
I am also concerned as when I checked the references it appears the verbiage used by the authors is very similar to the references cited such as the following example. Lines 39-48.
This section is reworded.Also reference #7 for this manuscript and Reference #7 in the Mushatt DM paper (ref #13) are the same paper.
Berbece AN, Richardson RM. Sustained low-efficiency dialysis in the ICU: cost, anticoagulation, and solute removal, Kidney Int, 2006, vol.70(pg.963-8)
As with continuous therapies, hemodynamic stability in SLED is better than in intermittent hemodialysis because of decreased ultrafiltration rates and runs for a longer time (Table 1). Conventional dialysis machines are used; thus, no additional equipment is needed. [7]. The dialyzers are also - inexpensive. A standard dialysate concentrate is used, rather than specialized dialysate and ultrafiltrate replacement solutions [7]. Furthermore, anticoagulation is not generally necessary for SLED [7]. Overall, the costs are considerably less than those for CRRT [8]. There are some disadvantages to SLED such as unfamiliarity with the modality, and hypophosphatemia [7]. Finally, one other disadvantage with SLED is that there is less removal of “middle molecules” in the size range 1000–10,000 Daltons compared to CRRT as the dialyzer membranes with SLED are generally less –permeable [13].
Reference 13 Mushatt DM, Mihm LB, Dreisbach AW, et al. Antibiotic dosing in slow extended daily dialysis. Clin Infect Dis. 2009;49:433- 437.
As with continuous therapies, hemodynamic stability in SLEDD is better than in intermittent hemodialysis because it is still less efficient and runs for a longer time. Conventional dialysis machines are used; thus, no additional equipment is needed, and the machine can be used for intermittent hemodialysis. The dialyzers are also relatively inexpensive. Routine dialysate concentrate is used, rather than specialized dialysate and ultrafiltrate replacement solutions. Overall, the operating expenses are considerably less than those for CVVH [6]. Furthermore, anticoagulation is not generally necessary for SLEDD [7]. There was less removal of “middle molecules” in the size range 1000–10,000 Daltons than with CVVH, because the membranes are generally less leaky, and this may be a disadvantage.
This section has been updated (both wording and references). We have updated the wording and reorganized it.
Throughout the manuscript the writing style uses “of” more than necessary for instance on line 99 “the clearance of the metabolites of the parent drug” the writing could easily be modified to a more succinct style.
Corrected- thank you for the comment
The following system does not appear to match what your reference states. Please correct.
Lines 36-39. Some differences between SLED and 37 other traditional continuous therapies include: clearances for small molecules are generally higher 38 per hour than they are in CRRT; remembering, however, that SLED is generally used for only 6–12 h 39 per day and thus overall there is less clearance [8].
Copied from the published reference.
“All systems offer the advantages of flexible timing of treatment and reduced costs; their ease of handling means that SLED is readily accepted by ICU staff. Prospective controlled studies have shown that SLED clears small solutes with an efficacy comparable to that of intermittent hemodialysis and continuous venovenous hemofiltration (even when the latter employs high rates of fluid substitution).” Fliser D, Kielstein JT. Technology insight: treatment of renal failure in the intensive care unit with extended dialysis. Nat Clin Pract Nephrol 2006; 2:32–9.
Those numbers 37 and 38 do not show up in my copy? Or are you referring to the line numbers. Needless to say- this section has been updated according to the references. This section has been reworded- according to our references used. We believe it is more clear.Line 97 remove the word both from both human studies.
correctedLine 500. Should a Heading of Conclusion be added here?
Conclusion heading added to line 663
References. Authors names misspelled.
Villay MA, Churchill MD, Mueller BA. Clinical review: Drug metabolism and nonrenal clearance in 578 acute kidney injury. Crit Care. 2008; 12(6): 235-244.
Authors misspelled. Should be Vilay AM, Churchwell MD, Mueller BA
Thank you - correctedReferences Formatting is not consistent.
For Example: Some authors the Initials are B.A or J.T. but other are BA or JT.
Example. Pieter Evenepoel, Bjorn K.I.Meijers, Bert R.M.Bammens, KristinVerbeke. Uremic toxins originating 568 from colonic microbialmetabolism. Kidney International(2009)76(Suppl 114), S12–S19.
References have been reformatted.
Reviewer 2 Report
Brown et al. reported a literature review of antimicrobial dosing in sustained low efficiency dialysis. This manuscript has been significantly improved after the revision. Please see minor comments below:
Line 46: Please correct the space before “Finally”
PK and PD principles:
Line 106: Did you want an extra line added here? This line is not consistent with the rest of manuscript format.
Line 118: Delete extra period after “…receiving RRT..”
Table 1: Did you want two extra empty rows in this table?
Reference list: Please use one citation style. For example, ref 6 and 7 does not have bolded publication year. Ref 14 journal is underlined instead of being italicized.
Author Response
Thank you to Reviewer#2 for the comments and suggestions. Please see below for our responses and the manuscript for the changes.
Reviewer #2- Comments and Suggestions for Authors
Brown et al. reported a literature review of antimicrobial dosing in sustained low efficiency dialysis. This manuscript has been significantly improved after the revision. Please see minor comments below:
Line 46: Please correct the space before “Finally”
Thank you - correctedPK and PD principles:
Line 106: Did you want an extra line added here? This line is not consistent with the rest of manuscript format.
Thank you- correctedLine 118: Delete extra period after “…receiving RRT..”
Thank you- correctedTable 1: Did you want two extra empty rows in this table?
Thank you-CorrectedReference list: Please use one citation style. For example, ref 6 and 7 does not have bolded publication year. Ref 14 journal is underlined instead of being italicized.
Thank you- corrected
Round 3
Reviewer 1 Report
Overall the revisions requested for this manuscript have been completed. The authors have completed a through review of the literature and summarized their findings in an objective manner. I do not see a need for additional peer-review
Only very minor grammar items to consider revising please see below (bolded)
Grammar errors
Line 46. With SLED has some disadvantages such as unfamiliarity with the modality, and hypophosphatemia
Line 58 With the uncertainty of oral absorption in critically ill patients, medications are usually administered intravenously. However, if oral medications are given, absorption may be decreased secondary to uremic toxins present .
Author Response
Thank you. My version attached does not include the word With (line 46)
Also there is no space after the last word in line 57.